# Overactivity of the Less Affected Side as a Possible Pattern of Asymmetry in Sitting in Patients Suffering from First-Time Ischemic Stroke—An Observational Study

**DOI:** 10.3390/brainsci13121716

**Published:** 2023-12-14

**Authors:** Agata Zdrowowicz-Doroz, Jakub Stolarski, Karolina Krzysztoń, Izabela Domitrz, Jan Kochanowski

**Affiliations:** Department of Neurology, The Faculty of Medicine and Dentistry, Medical University of Warsaw, Bielański Hospital, 80 Cegłowska St, 3rd Floor, 01-809 Warsaw, Poland; azdrowowicz@gmail.com (A.Z.-D.); karolina.krzyszton@wum.edu.pl (K.K.); izabela.domitrz@wum.edu.pl (I.D.); jan.kochanowski@wum.edu.pl (J.K.)

**Keywords:** stroke, rehabilitation, sitting position

## Abstract

It has been observed that in some people in the acute phase of ischemic stroke (IS) there is a tendency to shift the body weight towards the side more affected by the disease and a tendency to spontaneous movements of the upper and/or lower limbs (not covered by the neurological syndrome). The purposes of this study were: to define the kind of behavior observed, and to select symptoms which can predict its occurrence. Participants (n = 222) hospitalized due to first-time IS were assigned to three groups. A: 78 patients with no lateralization of the neurological syndrome (lateralization of the neurological syndrome—LoNS); B: 109 patients with LoNS; O+ group: 35 patients, who at the beginning of hospitalization presented, apart from LoNS, characteristic motor symptoms performed by the less affected side. Patients underwent therapy depending on the neurological symptoms. If the patient showed potential symptoms of a new phenomenon, overactivity of the less affected side (OLAS), a trial therapy (focused on this behavior) was used to confirm it. The predictive symptoms, selected among these from the index day, for the occurrence of OLAS in sitting were distinguished: asymmetry in supine posture and simple, repetitive movements of the nonparetic upper extremity.

## 1. Introduction

The observation of patients in the acute phase of ischemic stroke (IS) during the period from the beginning of hospitalization to the moment of standing up to vertical positions allowed us to select a group of patients whose spontaneous motor activity stood out from the typical clinical picture of hemiplegia syndrome [1]. It has been noticed that patients from this group have a tendency to shift the body weight toward the more affected side (MAS) in sitting which makes it more difficult for them to develop a stable upright position. An additional symptom noticed in these patients, which makes the re-education of postural function difficult, was the tendency to move the less affected upper extremity (LAUE) and/or the less affected lower extremity (LALE)—not covered by the neurological syndrome. These motor behaviors were also observed in the supine posture at the beginning of the patients’ hospitalization—see Appendix A. An additional interesting symptom presented by these patients was typical asymmetry in supine posture, pictured in Figure 1—the schema of the study. Based on clinical observations of patients, a study protocol was developed aiming to define the concept of overactivity of the less affected side (OLAS), characterize its typical symptoms, and determine whether, in line with the researchers’ hypothesis, observed movement behaviors in patients during the initial physiotherapeutic assessment could serve as predictive indicators for later therapeutic challenges in higher positions. To the best of the researchers’ knowledge, no work has been undertaken with such an approach regarding patients in the acute phase of ischemic stroke.

## 2. Materials and Methods

The study group consists of individuals who have experienced a first-time acute IS confirmed by magnetic resonance imaging or computed tomography. Patients with focal injury in the thalamus or/and cerebellum, with another neurological disease in their history, and with an incident of loss of consciousness during hospitalization were excluded from the study. The research was carried out in the years 2020–2021 at the Department of Neurology, Faculty of Medicine and Dentistry, Medical University of Warsaw, located at the Bielański Hospital. The research process collected data from 125 women (56%) and 97 men (44%). The average age of the patients was 71.82 ± 14.27 years and ranged from 29 to 99 years. The study included patients meeting the following criteria: conscious individuals with no prior history of any CNS (central nervous system) disorders. Patients diagnosed with hemorrhagic stroke, those experiencing hemorrhagic transformation of the stroke lesion during hospitalization, or those having an ischemic stroke as a recurrent event were excluded from the study. To ensure the most uniform neurological profile, patients with strokes localized in the cerebellum and thalamus were excluded from the study. Patients manifesting disturbances in consciousness or incidents of loss of consciousness during hospitalization were also excluded. Patients with coexisting neurological conditions in their medical history and those who experienced orthopedic injury (e.g., limb fractures, vertebral fractures, etc.) at the time of the ischemic stroke occurrence were excluded as well. Patients diagnosed with SARS-CoV-2 infection during hospitalization were also removed from the study. Patients meeting the aforementioned criteria were allocated to three groups, based on the presented neurological symptoms. Group A comprised 77 patients who, within the first day of physiotherapeutic assessment, exhibited neurological symptoms of slight or mild severity and achieved maximum scores on the prognostic Scandinavian Stroke Scale (SSSP). Group B included 109 patients who, within the first day of physiotherapeutic assessment, demonstrated clear lateralization of neurological symptoms involving muscle weakness, superficial and/or deep sensory disturbances, changes in muscle tone, and scored lower than the maximum on the prognostic Scandinavian Stroke Scale (SSSP). Group O+ encompassed 36 patients who scored lower than the maximum on the SSSP and, in addition to the lateralization of neurological symptoms related to muscle weakness, superficial and/or deep sensory disturbances, and changes in muscle tone, exhibited at least one characteristic movement typical of OLAS on the side unaffected by the neurological symptoms or showed typical asymmetry within the first day of physiotherapeutic assessment. The detailed characteristics defining patients within each group are presented in Figure 2.

### 2.1. Methodology

Patients were assessed on the following baseline parameters: ability to move the trunk muscles against the force of gravity according to Trunk Control Test (TCT) [2], ability to move the extremities according to the prognostic part of the Scandinavian Stroke Scale (SSSP) [3], deep sensation and tactile sensation of the affected limbs, muscle tone in the limbs, and hemineglect (tested by the bedside half-string test). Information about the length of hospitalization, possible death, occurrence of symptoms potentially predictive for OLAS—asymmetry in supine (AiS), tendency to move LAUE, tendency to move LALE—were assessed visually. Moreover, it was marked on which day of hospitalization the patient reached sitting and standing position for the first time. The listed parameters were assessed on the first day of physiotherapeutic assessment—the earliest day when the physiotherapist could examine the patient. If a patient was admitted to the ward on Friday evening, he was assessed on Monday morning. Weeks that included holidays and extended weekends were excluded from follow-up, and no patients were recruited for the study on those days. No other confounding variables were taken into consideration in the study.

### 2.2. Study Design

Patients underwent therapy depending on neurological symptoms. The A group underwent health education mainly regarding the possibility of performing physical exercises. The B group underwent standard therapy conducted in accordance with the guidelines of the European Stroke Organization, i.e., sensory stimulation in the case of sensory disorders, positioning on the more affected side to prevent shoulder subluxation, and strength training in the case of muscle weakness. The therapy was performed once a day, apart from Saturday and Sunday. Each session lasted 20–30 min, depending on the patient’s condition during the acute phase of the stroke. It is also essential to emphasize that the aforementioned physiotherapeutic procedures, including their principles of planning, individual tailoring to patients, and practical implementation, represent standard practices employed within the Clinic’s daily physiotherapeutic routines for hospitalized patients with acute stroke. This approach is endorsed by the European Stroke Organization and aligns with the International Classification of Diseases, 9th Revision, Clinical Modification (ICD-9-CM) for neurophysiological therapies: 93.3807, 93.3808. All post-stroke patients undergo the described therapy, while patients meeting the final inclusion criteria were subjected to statistical analysis. This action was indispensable for mitigating confounding factors and ensuring the creation of maximally homogeneous groups. If the patient demonstrated the symptoms of OLAS—weightbearing into the MAS and motor activity of the less affected extremities (LAE)—in a higher position than lying in the bed (at least sitting) a trial therapy (TT) was performed—described below. A TT based on the selected neurophysiological school techniques was created by the researchers. If such stimulation caused the patient to regain symmetry, he was designated as OLAS, and the patient was included in the statistical analysis as such. The described study schema is shown in Figure 1.

Patients in the O+ group underwent the therapy once a day (apart from Saturday and Sunday), which took from 20 to 30 min. The main difference between standard therapy and the trial therapy of patients with symptoms of OLAS was that in the initial phase of treatment the physiotherapeutic stimulation was focused on the less affected side (LAS). The therapy for OLAS patients followed specific principles derived from the NDT-Bobath method, validated by our clinical experience. The main principles of the therapy are as follows:During the acute phase of the condition, it is advisable to position the patient on the less affected side. The aim of this position is to encourage loading on the (overactive) less affected side;During the sitting phase, therapy is started by putting weight on the less affected side. This is achieved by adopting a seated posture and providing support on the upper limb of the less affected side. To increase the load on the less affected side, the more affected leg can be placed on the less affected lower limb, providing there are no contraindications such as advanced joint degeneration or hip joint endoprostheses;Additional stimulation of deep sensory receptors is applied by approximating the patient’s shoulder towards the supported limb of the less affected side;During the therapy, it is advised that the patient shifts their body weight towards the less affected side, including overcorrection beyond the midline in that direction. The position that loads the less affected side in overcorrection should be maintained until the patient relaxes and ceases to initiate position changes;The functional patterns of the patient’s more affected side are actively engaged, with a crossing of the body’s midline towards the less affected side. In cases where there is significant muscle weakness on the more affected side, passive movement is advised. Before such activity, it is recommended that the less affected side be loaded. In this context, an advantageous intervention for the patient involves attempting to mobilize them by leaning towards the less affected side. Further information on the trial therapy can be found in Figure 3.

## 3. Results

According to the study design, none of the patients from group A (see Table 1) had finally presented OLAS, and they were excluded from the further analysis. These patients suffered from a mild neurological syndrome, so further analysis comparing patients with and without OLAS could generate overestimated differences between them.

### 3.1. Baseline Data Analysis

Baseline neurological and selected symptoms typical for OLAS data have been compared and analyzed. The results are presented in Table 2. Data are shown as n (% of OLAS group/no OLAS group) unless otherwise noted. Odds ratio (OR) is noted for categorical variables.

### 3.2. Length of Hospitalization and Achievement of a Independent Sitting Position

As another step, the differences between the hospitalization time in patients with and without OLAS were compared. The hospitalization time was significantly longer in patients with OLAS (median = 14 days) compared to patients without OLAS (median = 10 days), *p* = 0.001. Moreover, it was noticed that some hospitalizations ended with death. There is a significant difference between the compared groups—the rate of deaths was also significantly higher among patients with OLAS (37% vs. 5.8% of patients without OLAS), OR = 9.4; *p* < 0.001. It shows that patients with OLAS suffered from a more severe condition, and they had a more than nine times lower chance to survive. The moment of achieving an active sitting and standing position between patients with OLAS and without OLAS was analyzed. Sitting was achieved by 29 subjects with OLAS (71%), compared to 97 subjects without OLAS (93%), *p* < 0.001. A visual comparison of the data in box plot form is shown in Figure 4 and Figure 5, presenting an empirical distribution of data showing proportional differences between the groups in the collected data on the time of verticalization sitting on the edge of the bed. The standing position was achieved by 15 subjects with OLAS (37%), compared to 92 subjects without OLAS (88.5%), *p* < 0.001. Data on the moment of reaching the sitting and standing position have been presented in Table 3.

Afterwards, data regarding the initial attempt of sitting were analyzed. Thirty-five patients showed a typical asymmetry with a shift of body weight to the MAS. It was checked which variants of symptoms occurred most often, and whether they differed significantly from each other. A summary of all observed combinations of symptoms is presented in Table 4. It should be noted that in four patients typical OLAS behavior (difficulty in weightbearing to the LAS) was observed while standing.

### 3.3. The Predictive Model Creation

In the next step, it was checked how many of the 35 patients who presented typical symptoms in sitting reacted positively to the TT. Twenty-nine patients (83%) who presented asymmetry in sitting or asymmetry in sitting with coexisting LAUE or/and LALE motor activity underwent the TT. From this group, 23 patients (79%) reacted positively to the TT; the remaining six patients (21%) did not achieve a positive effect, *p* = 0.005.

In the last step, in order to check whether baseline AiS or a tendency to repetitive movements of LAUE and/or LALE are a significant predictors of occurrence OLAS in sitting, a logistic regression analysis was performed. Only patients who achieved the sitting position were included in this analysis. Univariate analysis indicated that each of the considered predictors, analyzed separately, was statistically significant. Effect sizes noted for categorical crucial variables are shown in Table 5.

Then, an attempt was made to create a multidimensional model using the stepwise method and two baseline predictors were included in the final model. The quality of the final multivariate model was assessed using the chi-square likelihood ratio test (*p* < 0.001) and the Hoshmer–Lemeshow (goodness-of-fit) test (*p* > 0.999). The results indicate a good fit of the model. Negelkerke’s pseudo R2 of 44% was adequate. There was no significant correlation found among the predictors (VIF values—variance inflation factor was 1.18). The data presented above are summarized in Table 6.

## 4. Discussion

Observation of patients in the acute phase of IS in the period from the beginning of hospitalization to the moment of sitting and standing allowed us to select a group of patients whose spontaneous motor activity stood out in comparison to people presenting a typical clinical picture of hemiplegia. Based on the results of this research, inspired by the observation of the clinical condition of people in the acute phase of IS, the concept of OLAS was defined. In addition, the most typical symptoms of OLAS were characterized, and it was determined which neurological symptoms coexist with it. It was considered whether, in view of the above, the characteristic motor behaviors observed in patients on the first day of physiotherapeutic assessment could be predictive symptoms for later therapeutic problems in higher positions. To the best of the researchers’ knowledge, no work has yet been published with such an approach to the problem of patients in the acute phase of IS.

It was hypothesized that the group of patients presenting baseline symptoms of OLAS (additional movements of the LAE and/or AiS) would be the group that would also be characterized by asymmetry in high positions, and thus a diagnosis of OLAS can be made based on observation of the supine position. The results on the motor symptoms of OLAS collected from the analysis at the beginning of hospitalization showed that each symptom typical of OLAS from the index day, analyzed separately, was a significant predictor of the occurrence of OLAS in sitting. An attempt was also made to create a predictive model from more than one typical symptom of OLAS from the index day to check whether adding one significant symptom to another would improve the ability to predict that OLAS would actually occur in the sitting position. The conducted analysis gave the results that the coexistence of LAUE and LALE activity on the first day of physiotherapeutic assessment allows, with a certain probability, to predict the occurrence of OLAS in sitting. A review of the literature on the treatment of people with IS did not provide data with which the obtained result could be compared.

One of the hypotheses was the occurrence of characteristic, reproducible differences in sitting between patients with OLAS and people with a typical clinical picture of post-stroke hemiplegia. It was noticed that patients from the OLAS group have a tendency to shift the body weight to the MAS, which makes it more difficult for them to develop a stable sitting, and then standing, position. In addition, in sitting, patients also tended to move their LAE. To check the validity of this hypothesis, the way patients with OLAS presented themselves in a sitting position was examined. It was checked which behaviors in this position appear most often and whether the selected variants of motor behaviors are significantly different from others. Such action was needed to create the definition of OLAS. Of all the symptoms of OLAS in sitting presented by patients, three variants of motor behavior stand out from the statistical point of view. OLAS in the sitting position may be manifested not only by asymmetry of the trunk with a deviation towards the MAS. The coexistence of a tendency to move the LAE is also important. The mentioned symptoms are illustrated in Figure 6. Information has been found in the literature on the existence of other possible causes of sitting instability. These include, for example, antigravity muscle weakness, changes in muscle tone, hemineglect, or Pusher Syndrome—PS, which, apart from extension and abduction activity of the limbs on the LAS and passive resistance during posture correction, is characterized by tilting to the MAS. However, this is due to a specific reason—such patients incorrectly perceive the position of their body in the frontal plane, defining their position as correct (symmetrical) when it is tilted by 18–20 degrees to the MAS. PS patients (similar to OLAS patients) demonstrate abnormal activity of the LAS, and limited ability to change movement patterns. The clinical scale for contraversive pushing, in which the occurrence of the mentioned symptoms is evaluated (in contrast to OLAS) solely in the sitting and standing positions, as well as the presence of a lesion in the thalamus as a structural cause of PS mentioned in some sources, indicates the necessity to differentiate between OLAS and PS [4,5]. In this study, to ensure that PS was not the cause of sitting asymmetry, patients with a thalamic stroke were excluded from the research. Both OLAS and PS patients tend to shift body weight to the MAS. In patients with PS, pushing activity of the LAE typically leads to a fall on the MAS. In PS, the purpose of this behavior is to align the body relative to the vertical midline incorrectly perceived by the patient. On the other hand, in OLAS it is probably (as the researchers conclude from their clinical practice) due to the patient’s feelings—it is more comfortable for the patient to take the weight off the LAS, which is indicated by protesting motor behavior in the situation of hypercorrection and support on the LAE, e.g., in sitting. It should be emphasized, however, that patients with OLAS will allow positioning to support the LAS during therapy, which is much more difficult to achieve in the case of patients with PS. Patients with PS, when trying to hypercorrect to the LAS, react with strong resistance. In the treatment of a patient with the PS, many vertical reference points are needed, while in the case of OLAS, therapeutic activities focus on its extinction by stimulating weighting on the LAS.

Apart from the already mentioned PS, asymmetry in the sitting position may also be caused by weakening of the muscular strength of the limbs and trunk, decreased muscle tone, disorders of superficial and deep sensation, cerebellar syndrome, and hemineglect [6,7,8,9,10]. Among the assessed neurological symptoms in patients qualified for this study, there were many missing data caused by the lack of the verbal–logical contact with the patient which is necessary to conduct a full neurological examination. For this reason, superficial and deep sensation, or the presence of hemineglect, was impossible to assess in some cases. It is noteworthy that the group of patients diagnosed with OLAS consisted of patients with a significant severity of the neurological syndrome, similar to PS. Statistical analysis of data collected from patients in the second measurement period confirmed that patients with OLAS significantly more often suffered from decreased muscle tone in more affected lower and upper extremity severe paresis or paralysis of the limbs, sensory disturbances, and hemineglect. Therefore, it was considered whether the asymmetry occurring in OLAS may be caused by the above-mentioned disorders, or is even the result of several of them. A similar asymmetry to that observed in OLAS may be presented by patients with deficits in superficial and deep sensation, hemineglect, and more severe hemiparesis. Exactly the same symptoms appear to be significantly associated with the behavior of patients with OLAS. However, from the available literature, it can be learned that with such disorders, typical asymmetry (with a greater load on the MAS) is possible; however, it occurs less frequently than the opposite pattern—with the body weight shifted to the LAS. This would suggest that OLAS is a separate cause of sitting asymmetry. However, it cannot be ruled out that the above-mentioned disorders underlie the behavior of patients in OLAS. The causes of asymmetry observed in the seated posture among stroke survivors are outlined in Table 7.

At the study design stage, it was hypothesized that patients with OLAS symptoms from the first and second measurement time would also present symptoms of overactivity in standing. Standing data were collected from 37% of patients with OLAS. An interesting observation is that four patients (9.8%) showed signs of OLAS (strong transfer of body weight to the MAS and inability to weight more affected lower extremity) only in standing. The data collected from the patients did not show a significant relationship between the occurrence of OLAS in sitting and the pattern of weighting the more affected lower extremity in standing. This result indicates that patients diagnosed with OLAS in sitting do not necessarily have problems in standing. On the other hand, it cannot be said that patients with or without OLAS in sitting will not show the symptoms of OLAS in standing. An issue that requires comment is also the time when it was decided to facilitate the patient to a standing position. From the physiotherapy point of view, patients in the acute stage of IS, before facilitating the standing position, should first regain stability in sitting. As mentioned earlier, OLAS patients present the problem of trunk instability in sitting. It has been shown that patients with OLAS needed more time to be able to stand correctly and independently.

In purely theoretical terms, it is unclear whether the observed motor activities of the LAE in subjects with OLAS correspond to the hemispheric activity visible on fMRI. According to the literature, moving the more affected upper extremity during the acute phase of stroke triggers pathological stimulation in both hemispheres of the brain, while moving the less affected extremity does not have the same effect [11]. Could the continuous, repetitive movement of the LAE, which patients exhibited during their initial physiotherapy assessments, be directed towards “protecting” the damaged hemisphere in the event of significant damage? This study does not provide researchers with the means to answer this question. However, the mechanism of activation of the brain hemispheres after an ischemic stroke was considered when reflecting on physiotherapy methods. It was noted that movement of the more affected limbs induces pathological stimulation of both brain hemispheres, whereas movement of the less affected extremities (LAE) does not. Scientific reports provide information on physiotherapeutic interventions aimed at determining the impact of such activities on interhemispheric balance. In 2021, N. Salehi et al. published research attempting to answer whether increasing resistance exercises performed by LAUE could restore balance of interhemispheric activity visible in fMRI. Additionally, the study aimed to determine if patients undergoing this therapy would experience greater improvements in the motor functions of their more affected upper extremities compared to patients undergoing standard therapy. The study group exhibited a markedly greater enhancement in muscle strength and efficiency in comparison to the control group. The researchers identified improved interhemispheric balance as the mechanism responsible for enhancing strength in the more affected upper extremity [12]. It is remarkable that the use of the LAS does not induce any pathological pattern of interhemispheric activity and promotes proper activation, which is encouraging. However, as mentioned earlier, the relationship between interhemispheric brain activity and OLAS remains unknown—this issue requires further research.

In this research it has been assumed that the TT is a useful tool in the process of diagnosing OLAS. Patients reacting positively to this therapeutic intervention were diagnosed as OLAS. In everyday physiotherapeutic practice and in the functional diagnostics of each patient—also a patient after a stroke—the trial therapy should be used as a tool for programming an individual rehabilitation process. It allows one to assess whether the given techniques are acceptable by the patient, and constitute a movement challenge at an appropriate level of difficulty for him. The trial therapy is also a valuable element of the rehabilitation program evaluation. The main therapeutic rule based on neurophysiological schools is to reduce asymmetry by shifting weight onto the LAS. For the purposes of this study, the TT could be performed when the patient was able to actively, with little help, maintain a sitting position. The therapy was conducted on the patient’s bed. In this study, the TT was based on our theoretical knowledge, and our own observations and experience of working with patients. It has not been evaluated for effectiveness. It is worth highlighting that among the group of patients with OLAS, among all patients who qualified for the trial therapy (minimum asymmetry in the sitting position), 79% responded positively to the applied therapy. According to the study’s protocol, a positive response to the trial therapy confirmed that the observed motor symptoms were related to OLAS. Such a result suggests that the formulated principles of the trial therapy could serve as guidelines for conducting therapy in practice. The researchers took notice of a 2019 publication proposing therapy on a diagonally inclined surface aimed at addressing trunk instability in sitting. In this therapy, individuals with stroke in the subacute phase sat on a 10-degree inclined platform with their trunk rotated 45 degrees backward, initially favoring the unoccupied side. The therapy’s instruction for patients involved active or therapist-assisted tilting of the trunk towards the unoccupied side in a diagonal direction. Patients undergoing this rehabilitation demonstrated greater improvement in trunk stability compared to the control group who performed a similar therapeutic procedure on a flat platform. This therapy might also be applicable to individuals with other lower limb affected sides (OLAS); however, considering earlier reflections, positioning these patients on an inclined platform might require the opposite initial side weighting, tilting towards the initially unoccupied side [13].

Further analysis is required for a better understanding of OLAS. One of the stages of expanding knowledge about OLAS may be an attempt to modify the trial therapy, also in terms of its impact on overactivity observed in the supine. Furthermore, an intriguing and unexplored issue is the potential impact of OLAS on the long-term rehabilitation outcomes of patients, as well as the occurrence of OLAS in patients with hemorrhagic stroke and in those experiencing recurrent stroke incidents in their medical history.

### Limitations of the Study

The presented study is characterized by several limitations. One of them was the impaired verbal–logical communication with patients, due, among other factors, to aphasia and dementia syndrome. This resulted in a higher number of data gaps during patient assessments. Moreover, a considerable number of initially enrolled patients were lost due to infections. These infections included hospital-acquired bacterial and viral infections, with Clostridium difficile being the most common bacterial infection. Additionally, the study coincided with the SARS-CoV-2 virus epidemic. Patients in isolation were also excluded from the study. Hospitalization complicated by both bacterial and viral infections led to a worse condition in infected patients compared to those not infected, potentially influencing the assessed parameters in this study.

## 5. Conclusions

The study results allows to present the definition of OLAS as instability of the trunk in the sitting position with associated characteristic motor activities of less affected limbs. Moreover, predictive symptoms of OLAS were distinguished.

## Figures and Tables

**Figure 1 brainsci-13-01716-f001:**
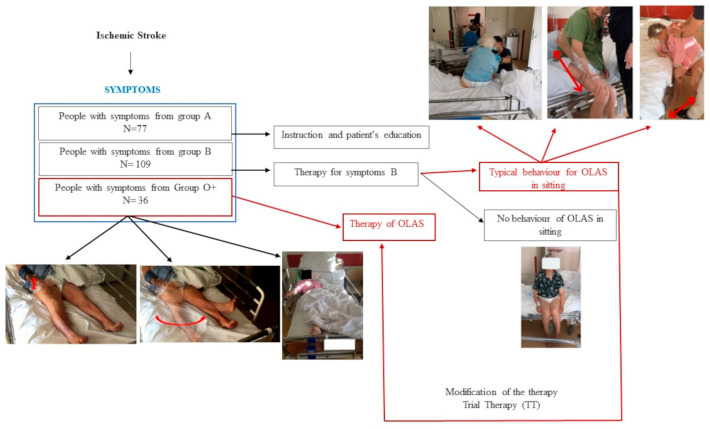
The schema of the study.

**Figure 2 brainsci-13-01716-f002:**
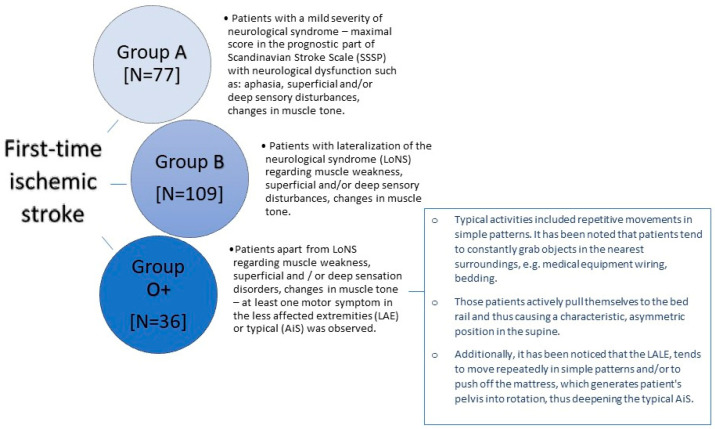
The characteristics of the groups.

**Figure 3 brainsci-13-01716-f003:**
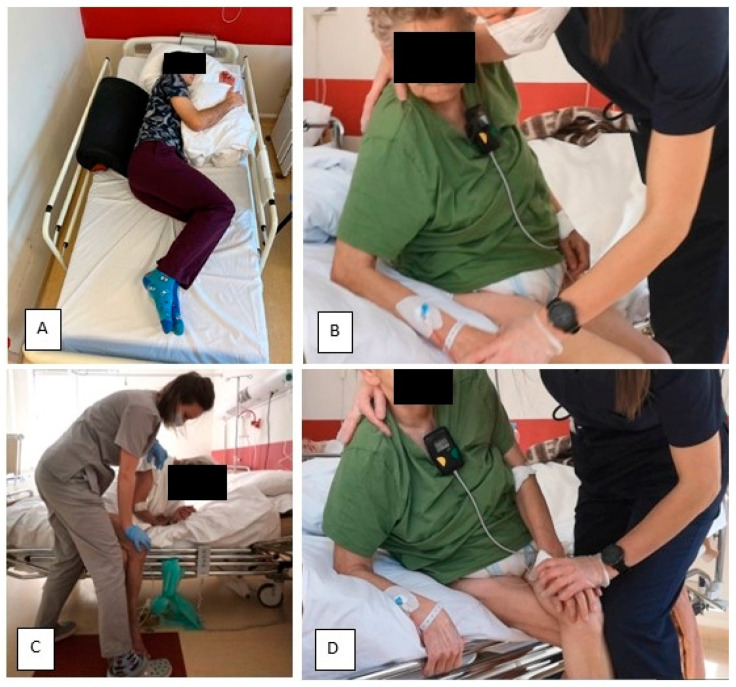
Trial Therapy. (**A**) In order to facilitate weight bearing to the overactive less affected side (LAS), patients have been laid on this side. (**B**) In sitting, therapy has been started by shifting weight to the LAS. The position with single support on the less affected upper extremity (LAUE) in sitting was useful. (**C**) To maximize the load on the LAS—if there were no contraindications—the patient’s more affected lower extremity was placed on the less affected lower extremity (LALE). Additional stimulation of deep sensory receptors was performed by applying approximation of the patient's shoulder towards the supported LAUE. This position was maintained until the patient became calm and stopped trying to change position. (**D**) Enforcement of the activity of the trunk in sitting using the more affected side (MAS) is beneficial, e.g., leaning forward.

**Figure 4 brainsci-13-01716-f004:**
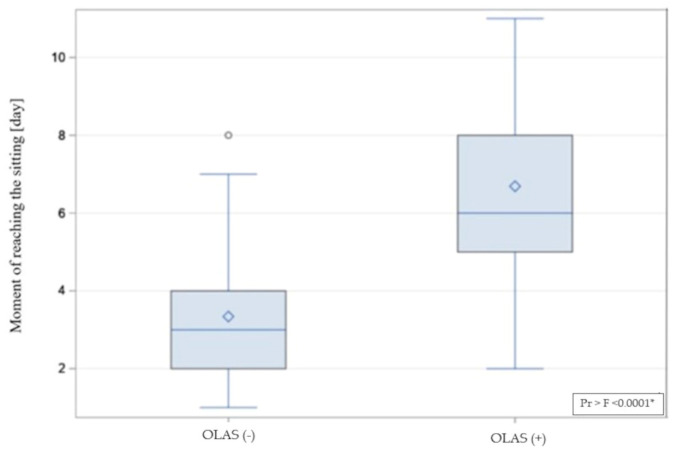
The visual comparison of the data on achieving a sitting position. * significance of correlation.

**Figure 5 brainsci-13-01716-f005:**
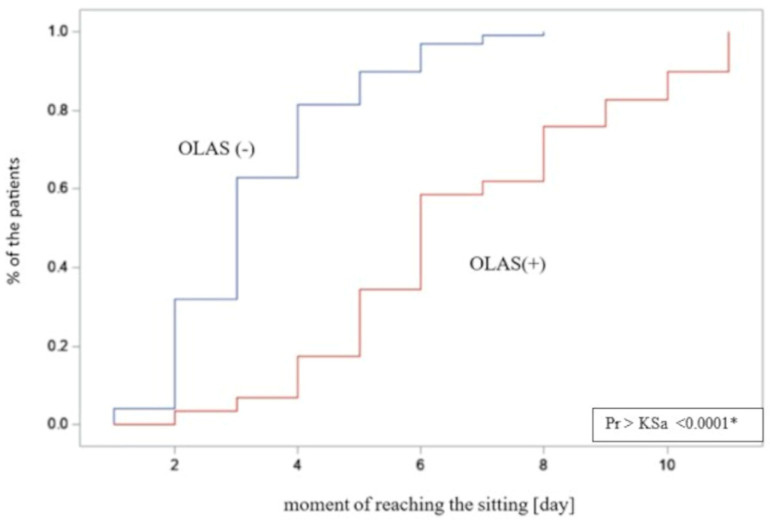
The empirical distribution of data on the moment of reaching free sitting on the edge of the bed by patients with and without OLAS. Patients diagnosed with OLAS achieved a seated position significantly later compared to patients without OLAS. * significance of correlation.

**Figure 6 brainsci-13-01716-f006:**
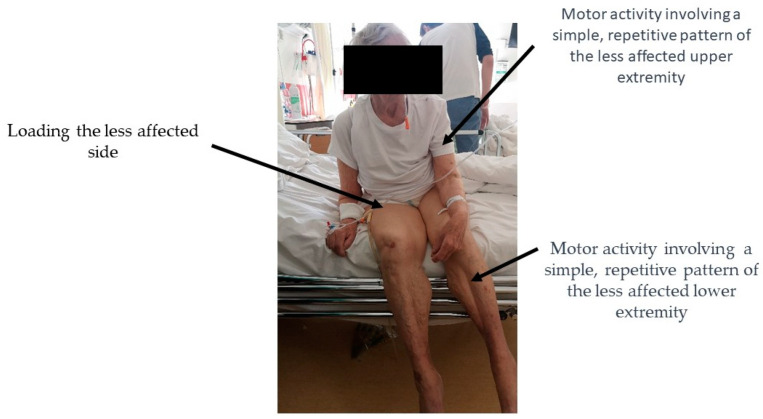
Based on the results of the statistical analysis, OLAS is defined as a distinctive form of trunk instability accompanied by activities in an extremity or extremities not covered by neurological symptoms (less affected).

**Table 1 brainsci-13-01716-t001:** The final number of patients with and without OLAS in each group.

Group	n	% of Group	% of Subgroup (A/B/O+)
A	77	34.7	
No OLAS	77	34.7	100.0
OLAS	0	0.0	0.0
B	109	49.1	
No OLAS	98	44.1	89.9
OLAS	11	5.0	10.1
O+	36	16.2	
No OLAS	6	2.7	16.7
OLAS	30	13.5	83.3

**Table 2 brainsci-13-01716-t002:** Comparison of participants’ baseline data between persons with and without OLAS. UE—upper extremity, LE—lower extremity.

Variable	OLAS	no OLAS	OR (95% CI)	*p*
sex, woman	10 (24.4)	37 (35.6)		0.069
sex, men	31 (75.6)	67 (64.4)		
age, median, (Q1; Q3)	82 (75.00; 89.00)	72 (64.50; 82.50)		<0.001
IS right hemisphere	15 (36.6)	57 (54.8)		0.208
IS left hemisphere	24 (63.4)	47 (45.2)		<0.001
SSSP, median (Q1; Q3)	10.00 (6.00; 14.00)	20.00 (17.00; 22.00)		<0.001
TCT, median (Q1; Q3)	0.00 (0.00; 24.00)	87.00 (49.00; 88.00)		<0.001
Muscle strength UE, median (Q1; Q3)	0.00 (0.00; 2.00)	5,00 (5.00; 6.00)		<0.001
Muscle strength LE, median (Q1; Q3)	2.00 (0.00; 4.00)	5.00 (4.00; 5.00)		<0.001
Hemineglect	7 (36.8)	2 (1.9)	27.13 (5.04; 145.9)	<0.001
Increased muscle tone, UE	8 (20.0)	21 (20.2)	0.99 (0.39; 2.46)	0.183
Decreased muscle tone, UE	29 (70.7)	37 (35.6)	4.37 (1.99; 9.58)	<0.001
Increased muscle tone, LE	9 (22.0)	13 (12.5)	1.97 (0.77; 5.043)	0.158
Increased muscle tone, LE	26 (63.4)	35 (33.6)	6.63 (3.04; 14.95)	<0,001
Tactile sensation disturbance, UE	30 (83.3)	33 (34.02)	9.69 (3.67; 25.63)	<0.001
Tactile sensation disturbance, LE	30 (78.9)	33 (18.4)	12.7 (4.78; 34.14)	<0.001
Deep sensation disturbance, UE	8 (61.5)	8 (9.2)	15.8 (4.17; 59.93)	<0.001
Deep sensation disturbance, LE	8 (61.5)	9 (10.47)	13.69 (3.68; 50.9)	<0.001
Asymmetry in supine posture	26 (63.4)	7 (6.7)	24.02 (8.87; 65.04)	<0.001
Tendency to move LAUE	28 (68.3)	6 (5.8)	35.18 (12.25; 100.98)	<0.001
Tendency to move LALE	17 (41.5)	3 (2.9)	23.84 (6.46; 87.97)	<0.001

**Table 3 brainsci-13-01716-t003:** The comparison of data on OLAS and non-OLAS patients’ achievement of sitting and standing. * significance of correlation.

Variable	OLAS	No OLAS	*p*
sitting, day, median (Q1; Q3)	6.00 (5.00; 8.00)	2.00 (2.00; 3.00)	<0.001 *
standing, day, median (Q1; Q3)	8.00 (6.50; 8.00)	4.00 (2.00; 5.00)	<0.001 *

**Table 4 brainsci-13-01716-t004:** The summary of data on all observed combinations of symptoms in sitting patients with and without OLAS. * significance of correlation.

Variable	OLAS	No OLAS	*p*
No asymmetry + no LAE activity	4 (14.8)	85 (87.6)	<0.001 *
Only asymmetry	5 (18.5)	5 (5.2)	<0.001 *
Asymmetry + LALE activity	2 (7.4)	0 (0.0)	
Asymmetry + LAUE activity	8 (29.6)	0 (0.0)	<0.001 *
Asymmetry + LAE activity	8 (29.6)	1 (1.0)	<0.001 *
No asymmetry + LALE activity	0 (0.0)	3 (3.1)	
No asymmetry + LAUE activity	0 (0.0)	3 (3.1)	

**Table 5 brainsci-13-01716-t005:** The score of one-dimensional logistic regression of baseline symptoms and occurrence of OLAS in sitting. * significance of correlation.

Predictor	OR	95% CI for OR	Se	Sp	PPV	*p*
Asymmetry	21.49	7.09–65.09	0.74	0.88	0.58	<0.001 *
Tendency to move LAUE	32.94	9.49–111.28	0.81	0.88	0.60	<0.001 *
Tendency to move LALE	21.38	4.29–05.53	0.80	0.82	0.31	<0.001 *

**Table 6 brainsci-13-01716-t006:** The score of multivariate prediction model of occurrence of OLAS in sitting. * significance of correlation.

Predictor	OR	95% CI for OR	*p*
Tendency to move LAUE	21.47	6.12–90.95	<0.001 *
Tendency to move LALE	4.44	0.62–40.18	0.147

**Table 7 brainsci-13-01716-t007:** Potential causes of trunk asymmetry in seated posture following stroke.

	Asymmetry in Sitting	
Causes of Asymmetry	1. Toward Less Affected Side	2. Toward More Affected Side
Hemiparesis	More often	Less often
Sensory deficits	More often	Less often
Pusher Syndrome	Absent	Present
Hemineglect	More often	Less often
Cerebellar syndrome	Absent	Present
OLAS	Absent	Present

## Data Availability

The data are not publicly available, due to the large amounts of sensitive data.

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
