# Peer review of "Overactivity of the Less Affected Side as a Possible Pattern of Asymmetry in Sitting in Patients Suffering from First-Time Ischemic Stroke—An Observational Study"

_brainsci, 2023, doi:10.3390/brainsci13121716_

Round 1

Reviewer 1 Report

Comments and Suggestions for Authors

The manuscript titled “Overactivity of the less affected side as a possible pattern of asymmetry in sitting in persons suffered from first-time ischemic stroke – an observative study”, presents a unique observation but needs refinement in terms of clarity, organization, and depth of discussion.

1.           The abstract lacks clarity and conciseness. It should succinctly introduce the problem, method, results, and conclusions. The transition between the abstract and the introduction is abrupt and could be improved for better flow.

2.           The introduction provides a comprehensive background but lacks a clear statement of the research question or hypothesis. It would benefit from a more explicit statement of the study's objective. It could be beneficial to include a brief review of existing literature on stroke-related motor behaviors, which would provide context for the novelty of the study.

3.           The statistical analysis could benefit from more detailed explanations, especially for non-specialist readers. Clarify the significance levels and effect sizes where applicable.

4.           While the discussion effectively interprets the study findings, it could delve deeper into potential implications for stroke rehabilitation and how these findings compare or contrast with existing literature. The link between observed motor activities in OLAS and hemispheric activity on fMRI is intriguing but speculative. Emphasize the need for further research to establish causation.

5.           The conclusion provides a concise summary, but it would benefit from reinforcing the clinical relevance of the findings and specifying potential avenues for future research and clinical application.

Comments on the Quality of English Language

Minor editing of English language required

Author Response

The manuscript titled “Overactivity of the less affected side as a possible pattern of asymmetry in sitting in persons suffered from first-time ischemic stroke – an observative study”, presents a unique observation but needs refinement in terms of clarity, organization, and depth of discussion.

 - Thank you for reading and formulating precise suggestions regarding improvements needed in the manuscript

  1. The abstract lacks clarity and conciseness. It should succinctly introduce the problem, method, results, and conclusions. The transition between the abstract and the introduction is abrupt and could be improved for better flow.

- The abstract has been revised for clarity and conciseness. Additionally, the introduction has been improved to ensure a smooth transition from the summary to the subsequent sections of the manuscript. The changes have been incorporated within lines 14-55.

  1. The introduction provides a comprehensive background but lacks a clear statement of the research question or hypothesis. It would benefit from a more explicit statement of the study's objective. It could be beneficial to include a brief review of existing literature on stroke-related motor behaviors, which would provide context for the novelty of the study.

    - As per the suggestions, the aims and the hypothesis of the study have been highlighted (lines 46-52). The review of literature concerning motor behaviors associated with stroke has been included in the discussion – lines 254-305)
  2. The statistical analysis could benefit from more detailed explanations, especially for non-specialist readers. Clarify the significance levels and effect sizes where applicable.

- The collected data underwent statistical analysis using 'R' software, version 4.0.5. Variables were presented using descriptive statistics appropriate for the measurement scale: qualitative variables were presented using absolute frequency 'n' and percentage frequency, while quantitative variables were presented as mean and standard deviation or as median and quartiles (first and third), depending on the distribution of the data. Normality of distributions was assessed using the Shapiro-Wilk test, visual inspection of histograms, and assessment of skewness and kurtosis indicators. Comparison between patients with OLAS and patients without OLAS was conducted using the chi-square test or Fisher's test for qualitative variables, and the Welch's t-test for independent measures or Mann-Whitney U test for quantitative variables, as appropriate to meet assumptions. Additionally, the odds ratio (OR) between both groups was calculated with a 95% confidence interval (CI). Logistic regression analysis was also performed to determine predictors of asymmetry in sitting and standing, as well as OLAS. Both univariate and stepwise multivariate analyses were employed for variable selection. Survival curves were plotted using the Kaplan-Meier method. Survival comparison between groups of patients with OLAS versus those without OLAS was conducted using the log-rank chi-square test. Statistical significance was set at α = 0.05 in the computations.

  1. While the discussion effectively interprets the study findings, it could delve deeper into potential implications for stroke rehabilitation and how these findings compare or contrast with existing literature. The link between observed motor activities in OLAS and hemispheric activity on fMRI is intriguing but speculative. Emphasize the need for further research to establish causation.

- Following the suggestions, the potential implications for rehabilitation were further elaborated. Additionally, a study investigating the effect of therapy on the inclined support surface concerning the improvement of sitting balance has been cited. These modifications are reflected in lines 359-375.

Furthermore, in the discussion, emphasis was placed on the need for further research to confirm the causal relationship between interhemispheric activity and motor behaviors characteristic of OLAS (lines 345-346).

  1. The conclusion provides a concise summary, but it would benefit from reinforcing the clinical relevance of the findings and specifying potential avenues for future research and clinical application.

 - The clinical significance of the research findings has been highlighted in the discussion (lines 359-364), particularly regarding the application of the experimental therapy in the daily practice with patients hospitalized in the stroke unit. Furthermore, potential directions for further research were outlined and described in lines 379-382.

Reviewer 2 Report

Comments and Suggestions for Authors

The manuscript presents an observational study focused on a novel phenomenon termed overactivity of the less affected side (OLAS) in patients during the acute phase of a first-time ischemic stroke, a subject that appears to be underrepresented in current research. The study's design and its potential implications for stroke rehabilitation are commendable. The study is comprehensive, with a well-structured approach to defining and characterizing OLAS. The use of substantial sample size, validated assessment tools, and the development of a predictive model for OLAS are strengths of the study. The distinction of OLAS from similar conditions, such as Pusher Syndrome, and the implications for rehabilitation strategies are particularly noteworthy. However, there are areas where the manuscript could be improved to enhance clarity, robustness, and the overall impact of the findings.

Improvements suggestions:

1.  Provide a more detailed description of the criteria used for patient categorization into the three groups (A, B, O+).

2.  Elucidate the specific neurophysiological techniques selected for trial therapy, and justify their choice based on existing literature or preliminary findings.

3.  Include a more detailed explanation of the statistical methods, particularly the rationale behind the choice of odds ratios and the logistic regression model.

4.  Discuss any potential confounding variables and how they were accounted for in the analysis.

5.  Provide a more comprehensive interpretation of the statistical results, including a discussion on the clinical relevance of the statistical significance levels observed.

6.  Explore the potential impact of OLAS on long-term rehabilitation outcomes, which is currently not addressed.

7.  Address the potential limitations of the study, such as the observational design and its implications for the causality of findings.

8.  Discuss the generalizability of the results to broader stroke populations, considering the study's specific inclusion and exclusion criteria.

9.  Propose how the findings of the study could inform future research, particularly in exploring the underlying mechanisms of OLAS.

10.    Discuss how the findings can be translated into practical clinical interventions for stroke rehabilitation.

11.    Incorporate a more detailed discussion on how the study's findings align with or differ from the existing body of literature on post-stroke motor behavior.

12.    Consider including flow diagrams or additional visual aids that outline the patient's journey from assessment through to treatment and follow-up.

Author Response

The manuscript presents an observational study focused on a novel phenomenon termed overactivity of the less affected side (OLAS) in patients during the acute phase of a first-time ischemic stroke, a subject that appears to be underrepresented in current research. The study's design and its potential implications for stroke rehabilitation are commendable. The study is comprehensive, with a well-structured approach to defining and characterizing OLAS. The use of substantial sample size, validated assessment tools, and the development of a predictive model for OLAS are strengths of the study. The distinction of OLAS from similar conditions, such as Pusher Syndrome, and the implications for rehabilitation strategies are particularly noteworthy. However, there are areas where the manuscript could be improved to enhance clarity, robustness, and the overall impact of the findings.

- Thank you for reading and formulating precise suggestions regarding improvements needed in the manuscript

1.  Provide a more detailed description of the criteria used for patient categorization into the three groups (A, B, O+).

  • In the Materials and Methods section, the description of patient allocation to groups was enriched with inclusion and exclusion criteria. Additionally, within the manuscript, emphasis was placed on presenting detailed information about patient allocation to respective groups graphically in Figure 1.
  1. Elucidate the specific neurophysiological techniques selected for trial therapy, and justify their choice based on existing literature or preliminary findings.

- Further details explaining the selection of techniques employed in the trial therapy have been added and are now included in lines 125-129.

  1. Include a more detailed explanation of the statistical methods, particularly the rationale behind the choice of odds ratios and the logistic regression model.
  • Logistic regression method was employed when the dependent variable takes only two values—whether it was present or not. For instance, the tendency to move LAUE in a repetitive pattern.
  1. Discuss any potential confounding variables and how they were accounted for in the analysis.

    - Additional information has been added in lines 103-104, specifying that other confounding factors apart from those described in Section 2.1 Methodology were not taken into account.
  2. Provide a more comprehensive interpretation of the statistical results, including a discussion on the clinical relevance of the statistical significance levels observed.
  • In the discussion, within lines 359-364, considerations regarding the results of the statistical analysis concerning the experimental therapy were added, highlighting the significance of these findings for clinical physiotherapy practice.
  1. Explore the potential impact of OLAS on long-term rehabilitation outcomes, which is currently not addressed.

- In the discussion, within lines 379-382, the need for further research was highlighted to determine the impact of OLAS on long-term progress in patient rehabilitation.

  1. Address the potential limitations of the study, such as the observational design and its implications for the causality of findings.

- Following the suggestions, the manuscript has been supplemented with a description of potential study limitations. These have been outlined in Section 4.1 'Limitations of the study' spanning lines 384-393.

  1. Discuss the generalizability of the results to broader stroke populations, considering the study's specific inclusion and exclusion criteria.
  • In the discussion, within lines 379-382, the need for further research was highlighted to investigate OLAS in patients with hemorrhagic stroke and those who have experienced more than one stroke in their medical history was suggested.
  1. Propose how the findings of the study could inform future research, particularly in exploring the underlying mechanisms of OLAS.
  • Following the suggestions, in lines 345-346 of the discussion, it was emphasized that the mechanism of interhemispheric activity, which theoretically could be related to OLAS, requires further investigation
  1. Discuss how the findings can be translated into practical clinical interventions for stroke rehabilitation.
  • In line with the suggestion, practical implications of the research results for clinical physiotherapy practice have been incorporated in lines 359-364.
  1. Incorporate a more detailed discussion on how the study's findings align with or differ from the existing body of literature on post-stroke motor behavior.

 The discussion on this topic has been enriched with an additional literature reference. The added text can be found within lines 364-375.

  1. Consider including flow diagrams or additional visual aids that outline the patient's journey from assessment through to treatment and follow-up.
  • Researchers made every effort to ensure transparency regarding the patient's journey during hospitalization, from allocation to respective groups to the verification of whether the patient presented OLAS or not. As a result, this entire process has been depicted in Figure 2.

Round 2

Reviewer 2 Report

Comments and Suggestions for Authors

Considering the answers and improvements made by authors after the first review step, I conclude for acceptance in the present form.

Author Response

Thank you for the thorough analysis of our manuscript and for the insightful and valuable comments.